# Immediate Adverse Events Following COVID-19 Vaccination in Australian Pharmacies: A Retrospective Review

**DOI:** 10.3390/vaccines10122041

**Published:** 2022-11-29

**Authors:** Alexander T. Gallo, Lisa Scanlon, Jade Clifford, Lawson Patten-Williams, Lachlan Tweedie, Dani Li, Sandra M. Salter

**Affiliations:** 1Discipline of Pharmacy, School of Allied Health, University of Western Australia, Crawley 6009, Australia; 2MedAdvisor International, Camberwell 3124, Australia

**Keywords:** COVID-19, vaccination, adverse event following immunization, pharmacist, pharmacy, anaphylactic reaction, anaphylaxis, syncope, seizure, convulsion

## Abstract

Background: Four COVID-19 vaccines are approved for use in Australia: Pfizer-BioNTech BNT162b2 (Comirnaty), AstraZeneca ChAdOx1 (Vaxzevria), Moderna mRNA-1273 (Spikevax), and Novavax NVX-CoV2373 (Nuvaxovid). We sought to examine the type and management of immediate adverse events following immunisation (I-AEFI) after COVID-19 vaccination. Methods: Retrospective review of I-AEFI recorded between July 2021 and June 2022 in 314 community pharmacies in Australia. Results: I-AEFI were recorded in 0.05% (n = 526/977,559) of all COVID-19 vaccinations (highest: AstraZeneca (n = 173/161,857; 0.11%); lowest: Pfizer (n = 50/258,606; 0.02%)). The most common reactions were: (1) syncope, after the first dose of AstraZeneca (n = 105/67,907; 0.15%), Moderna (n = 156/108,339; 0.14%), and Pfizer (n = 22/16,287; 0.14%); and (2) Nausea/vomiting after the first dose of Pfizer (n = 9/16,287; 0.06%), Moderna (n = 55/108,339; 0.05%), and AstraZeneca (n = 31/67,907; 0.05%) vaccines. A total of 23 anaphylactic reactions were recorded (n = 23/977,559; 0.002%), and 59 additional I-AEFI were identified using MedDRA^®^ terminology. Pharmacists primarily managed syncope by laying the patient down (n = 227/342; 66.4%); nausea/vomiting was managed primarily by laying the patient down (n = 62/126; 49.2%), giving water (n = 38/126; 30.2%), or monitoring in the pharmacy (n = 29/126; 23.0%); anaphylactic reaction was treated with adrenaline (n = 18/23; 78.3%) and n = 13/23 (56.5%) anaphylactic reactions were treated with the combination of: administered adrenaline, called ambulance, and laid patient down. Conclusion: The most commonly recorded I-AEFI was syncope after COVID-19 vaccination in pharmacy; I-AEFI are similar to those previously reported. Pharmacists identified and managed serious and non-serious I-AEFI appropriately and comprehensively.

## 1. Introduction

In December 2019, a new species of coronavirus emerged, which was linked to a seafood and wet animal wholesale market in Wuhan, China. The virus was named severe acute respiratory syndrome coronavirus 2 (SARS-CoV-2), which caused the coronavirus disease 2019 (COVID-19); after widespread infection, COVID-19 was declared, by the World Health Organization, a global pandemic [1,2,3]. Despite the usually lengthy process of vaccine development, remarkably, several COVID-19 vaccine candidates began clinical trials in less than six months with provisional registration by the Therapeutic Goods Administration (TGA) in Australia approximately 10 months after the pandemic was declared [4,5]. As the pandemic continued, new variants of COVID-19 emerged, leading to increases in breakthrough infections. This made booster doses particularly important in the vaccine rollout [6,7].

The COVID-19 vaccination rollout strategy by the Australian Government was to vaccinate individuals in five phases (1a, 1b, 2a, 2b, and 3), vaccinating those at highest risk of infection, complications, or death in the earlier phases [8], with vaccination of the general population by community pharmacists from phase 2b [9]. Importantly, when community pharmacists entered the Australian rollout, pharmacies were the only destination from which Moderna’s mRNA-1273 (Spikevax^®^) vaccines could be obtained, and pharmacists played a critical role in boosting COVID-19 vaccination rates.

Whole-of-population vaccination, coupled with public concerns around the speed of vaccine development and heightened vaccine hesitancy [10,11], meant vaccine safety surveillance was vital for the COVID-19 vaccine rollout. In Australia, established passive and active vaccine safety surveillance systems monitor adverse events following immunisation (AEFI) for all vaccines, including COVID-19 vaccines. However, these systems largely capture AEFI in the days following vaccination and are not structured to systematically capture AEFI occurring in the 15–30 min monitoring period immediately post vaccination.

Given the important role played in the COVID-19 vaccine rollout by pharmacists, and in light of early concerns about anaphylaxis to COVID-19 vaccines [12,13], we implemented a system for pharmacists in Australia to record immediate-AEFI (I-AEFI; those occurring within 30 min) post-COVID-19 vaccination, including how the reactions were managed. Limited studies have examined I-AEFI to COVID-19 vaccination [14,15,16] and currently, there is no data about I-AEFI to COVID-19 vaccines administered by pharmacists globally. Accordingly, we conducted a retrospective review of I-AEFI records for COVID-19 vaccines administered in Australian community pharmacies. The aim was to determine the type of I-AEFI experienced following COVID-19 vaccination in pharmacies and their management by pharmacists.

## 2. Methods

### 2.1. Design

Retrospective review of pharmacist records of I-AEFI after COVID-19 vaccinations in Australian community pharmacies.

### 2.2. Setting

All Australian community pharmacies using the MedAdvisor PlusOne software (MedAdvisor International Pty Ltd., Camberwell, VIC, Australia) to record COVID-19 vaccination encounters had access to an I-AEFI recording form. Pharmacies in which the I-AEFI recording form was used at least once between July 2021 to June 2022 were included in the study.

### 2.3. I-AEFI Recording Form

The I-AEFI recording form was developed by the researchers. The form was located within the MedAdvisor PlusOne software used to record vaccination encounters and visible only for COVID-19 vaccinations. In line with best practice, pharmacists were instructed to record any I-AEFI that occurred in the 15–30 min monitoring period after vaccination to provide a clinical record for the individual patient, which could inform future vaccination plans. The form was designed for pharmacists to record pre-defined I-AEFI that they might encounter (anaphylaxis, fainting, and nausea or vomiting), and management that they might provide (administered adrenaline, called ambulance, and laid patient down). The pre-defined I-AEFI map to the MedDRA^®^ terms anaphylactic reaction, syncope, and nausea/vomiting, respectively. MedDRA^®^ is the Medical Dictionary for Regulatory Activities terminology, which is the international medical terminology developed under the auspice of the International Council for Harmonisation of Technical Requirements for Pharmaceuticals for Human Use. Free-text options for ‘other’ I-AEFI and ‘other’ management were also available. Pharmacists completed the form by selecting either the pre-defined I-AEFI and management types through a multi-select tick-box, and/or by typing additional details in the fields for ‘other’ (Appendix A).

### 2.4. Participants

Participants were individuals five years and over who received a COVID-19 vaccination in a pharmacy using MedAdvisor PlusOne software and experienced an I-AEFI, which was recorded by the pharmacist using the I-AEFI recording form. As per the Australian COVID-19 vaccine rollout, the vaccinations available in community pharmacies were ChAdOx1 nCoV-19 (AZD1222) (Vaxzevria^®^ [AstraZeneca]), BioNTech BNT162b2 (Comirnaty^®^ [Pfizer, and Pfizer Paediatric]), mRNA-1273 (Spikevax^®^ [Moderna and Moderna Paediatric]), and NVX-CoV2373 (Nuvaxovid^®^ [Novavax]); hereafter termed by the manufacturer name. Pharmacies progressively offered the vaccines in accordance with the Australian Government policy and vaccine availability; broadly, AstraZeneca was available from July 2021, Moderna from September 2021, Pfizer from November 2021, paediatric formulations from January 2022, and Novavax from February 2022. Participants were required to stay in the pharmacy for 15–30 min post-vaccination, as per the Pharmacist Vaccination Code and Australian Immunisation Handbook (AIH) to monitor for any I-AEFI [17,18].

### 2.5. Variables

Patient demographics and vaccination details were sourced from the associated pharmacy vaccination encounter record in MedAdvisor PlusOne. Patient demographics included: age, sex, history of an adrenaline autoinjector dispensed, and vaccinating pharmacy location by Modified Monash Model classification (MMM) [19]. The MMM classifies locations based on whether they are a city, rural, remote, or very remote. The classifications range from 1–7 where 1 represents a metropolitan area and 7 represents a very remote community.

Vaccination details included vaccination date and COVID-19 vaccine brand and dose. Pharmacists administered a range of COVID-19 vaccine doses: first, second, third primary (for immunocompromised individuals), and booster; in this study we considered third primary or booster as ‘third+’.

Types of I-AEFI and management were sourced from the I-AEFI recording form. All records were screened and coded as follows:

Pre-defined I-AEFI (anaphylactic reaction, syncope, and nausea/vomiting), were those selected by pharmacists using the tick-box function in the I-AEFI recording form or written in free text, which mapped, in accordance with MedDRA^®^, to the tick-box terms. All ‘other’ (free text) I-AEFI were coded in duplicate based on MedDRA’s^®^ lowest level terms (LLT) and/or preferred terms (PT).

Pre-defined management were those actions (administered adrenaline, called ambulance, and laid patient down) selected by pharmacists using the tick-box function in the I-AEFI recording form or written in free text and thematically aligned with those actions. All ‘other’ (free text) management actions were coded in duplicate using an inductive thematic approach, with each distinct action creating a new management term.

Medical referral was considered to be any instance where a pharmacist referred, or where the patient stated an intention to see a medical professional, as part of management of an I-AEFI; for example, where the patient booked an appointment with a general practitioner (GP) or stated they would attend the emergency department. Medical attendance was considered to be any instance where a pharmacist recorded that the patient was seen by a medical professional; for example, where the pharmacist physically transferred the patient to a GP, or an ambulance attended. Accordingly, medical referral was when the pharmacist instructed the patient to seek medical attention; medical attendance was when the pharmacist had confirmed that the patient sought medical attention. The distinction was made between referral and attendance as it was not possible to confirm whether a patient who was *referred* actually followed through with *attendance.*

### 2.6. Analysis

In some cases, pharmacists noted reports of AEFI, made by patients on subsequent visits to the pharmacy, in the ‘other’ free text fields. These represented reactions that had occurred after the patient had left the pharmacy, and thus could not be considered I-AEFI. Accordingly, non-immediate AEFI (for example, where patients later reported a reaction that occurred overnight or in the days following vaccination), which were clearly noted by the pharmacist as not occurring in the pharmacy, were excluded from analysis. Any reaction noted by the pharmacist to have occurred during the 15–30 min monitoring period was included as an I-AEFI. The remaining records were assumed to represent AEFI that occurred during the monitoring period, as per the stated purpose of the form, and thus were considered to be I-AEFI.

All data were analysed descriptively using Excel (v2210; 2022; Microsoft; Redmond WA, USA), and SPSS (v28; 2021; International Business Machines Corporation; Armonk NY, USA). For pre-defined I-AEFI and those coded as seizure, type of I-AEFI were reported as the count, proportion of the total number of COVID-19 vaccines administered, and as an approximate rate per 1,000,000 COVID-19 vaccines administered by brand and dose. Approximate rates were calculated based on the formula below:$$\frac{n{\left(IAEFI\right)}_{specific}}{n{\left(vaccinations\;administered\right)}_{brand\;and\;dose}}\times 1,000,000$$

All other types of I-AEFI are reported as counts of MedDRA^®^ PT or LLT by COVID-19 vaccine and dose.

Management actions are reported descriptively for the pre-defined I-AEFI and seizure. Specifically, for syncope and seizure, we considered management based on the pharmacy’s geographic location by MMM classification, simplified to three categories: metropolitan or regional (MMM 1–2), which were grouped as they typically have reasonable access emergency services (i.e., in a major city or located within 20 km road distance to a town with a population greater than 50,000); rural (MMM 3–5); and remote (MMM 6–7). Specifically, for anaphylactic reaction, we considered management based on the pharmacy’s geographic location and, additionally, in terms of whether management met recognised ‘correct’ actions. Geographic location was based on the simplified MMM categories as described above. Correct management of anaphylactic reaction was based on the Australasian Society of Clinical Immunology and Allergy (ASCIA) recommendations for management of acute anaphylaxis [20]; correct management was considered to have been undertaken if the pharmacist had administered adrenaline, called an ambulance, and laid the patient down. We also considered management of pre-defined I-AEFI and seizure in terms of medical referral and medical attendance.

### 2.7. Ethics

This study was approved by the University of Western Australia Human Research Ethics Committee (2022/ET000316).

## 3. Results

A total of 2248 pharmacies had access to the I-AEFI recording form through MedAdvisor PlusOne software. Pharmacists from 314 (14%) pharmacies used the I-AEFI recording form at least once during the study period, making them eligible for inclusion in the study. Pharmacists in these pharmacies administered 977,559 COVID-19 vaccines. Of these, a total of 526 individuals experienced one or more I-AEFI after a COVID-19 vaccination.

Vaccines administered were AstraZeneca (n = 161,857), Moderna (n = 513,974), Moderna Paediatric (n = 560), Novavax (n = 11,967), Pfizer (n = 258,606), and Pfizer Paediatric (n = 30,595). For those in whom an I-AEFI was recorded (n = 526), the median age was 27.0 years (range: 6–91; IQR: 19.0), and the majority were male (n = 279/526; 53.0%). Two participants had a history of adrenaline autoinjector being dispensed. Most participants (n = 405/570; 77.0%) were vaccinated in metropolitan and regional areas (Modified Monash Model [MMM] classification 1–2) and were receiving their first dose (n = 397/526; 75.5%) of a COVID-19 vaccine (Table 1).

### 3.1. Immediate-AEFI

I-AEFI were recorded in 0.05% (n = 526/977,559) of all COVID-19 vaccinations. Participants receiving AstraZeneca vaccines had the highest proportion of any I-AEFI recorded (n = 173/161,857; 0.11%), while the lowest was Pfizer (n = 50/258,606; 0.02%) (Table 1). Of the pre-defined I-AEFI (anaphylactic reaction, syncope, and nausea/vomiting), syncope occurred most frequently, and was most commonly recorded following the first dose of AstraZeneca (n = 105/67,907; 0.15%), Moderna (n = 156/108,339; 0.14%), and Pfizer (n = 22/16,287; 0.14%) vaccines. Nausea/vomiting was most commonly recorded following the first dose of Pfizer (n = 9/16,287; 0.06%), Moderna (n = 55/108,339; 0.05%), and AstraZeneca (n = 31/67,907; 0.05%) vaccines. A total of 23 cases of anaphylactic reactions were recorded (n = 23/977,559; 0.002%). Anaphylactic reaction was most commonly recorded after the first dose of AstraZeneca (n = 6/67,907; 0.01%) and Moderna (n = 9/108,339; 0.01%) vaccines. Of the two participants with a history of adrenaline autoinjector being dispensed, one experienced anaphylaxis and the other experienced nausea/vomiting. Seizure most commonly occurred following the first dose of Moderna (n = 12/108,339; 0.01%) and Pfizer (n = 2/16,287; 0.01%). No I-AEFI were observed after the Novavax vaccination, aside from syncope after the first dose (n = 4/4665; 0.09%) (Table 2).

When considering approximate rates of I-AEFI, syncope, nausea/vomiting, and seizure were most commonly reported after the first dose of all vaccines except for Novavax, where no I-AEFI were recorded for syncope, nausea/vomiting, or seizure. Interestingly, approximate rates of syncope, nausea/vomiting, and anaphylactic reaction were generally lower following the second or third+ vaccination compared to the first. The largest difference between approximate rates was observed for syncope following the first (1546.23/10^6^) and second (111.21/10^6^) doses of the AstraZeneca vaccine. The rates of anaphylactic reaction varied for all vaccines with the AstraZeneca vaccine showing the highest approximate rate of anaphylactic reaction across all doses (43.25/10^6^) and the first dose (88.36/10^6^); however, the rate of anaphylactic reaction following the first AstraZeneca dose was comparable to the first Moderna dose (83.07/10^6^).

I-AEFI recorded as ‘other’ in free text, identified 59 additional types of reactions when mapped to MedDRA^®^ terminology. The most common were hyperhidrosis, injection site haemorrhage (bleeding), paraesthesia, and pain in extremity (Table 3). A table of the count of other I-AEFI by brand and dose can be found in Appendix A.

### 3.2. Management

Pharmacists principally managed syncope (n = 342) by laying the patient down (n = 227/342; 66.4%), calling an ambulance (n = 17/342; 5.0%), or both (n = 52/342; 15.2%). One case was managed with adrenaline and being laid down while two cases were managed with adrenaline, being laid down, and calling an ambulance. Other common management actions for syncope (of which multiple could be undertaken) included giving water (n = 64/342; 18.7%), monitoring in the pharmacy until symptoms resolved (n = 51/342; 14.9%), checking blood pressure (n = 29/342; 8.5%), giving glucose, lollipop, or chocolate (n = 28/342; 8.2%), checking pulse (n = 26/342; 7.6%), sitting the patient down (n = 20/342; 5.8%), advising the patient to self-monitor at home for worsening reaction (n = 11/342; 3.2%), giving a cold compress (n = 10/342; 2.9%), and elevating legs (n = 8/342; 2.3%). A total n = 4/342 (1.2%) of syncope reactions were medically referred, while n = 82/342 (24.1%) were medically attended (Table 4).

Pre-defined management of nausea/vomiting (n = 126) most commonly included laying the patient down (n = 62/126; 49.2%), calling an ambulance (n = 11/126; 8.7%), or both (11/126; 8.7%). Other management actions (of which multiple could be undertaken) included giving water (n = 38/126; 30.2%), monitoring in the pharmacy until symptoms resolved (n = 29/126; 23.0%), giving glucose, lollipop, or chocolate (n = 13/126; 10.3%), sitting the patient down (n = 11/126; 8.7%), giving a cold compress (10/126; 7.9%), advising the patient to self-monitor at home for worsening reaction (n = 8/126; 6.3%), checking pulse (n = 7/126; 5.6%), and monitoring breathing (n = 5/126; 4.0%). None of the nausea/vomiting reactions were medically referred, while n = 34/126 (26.9%) were medically attended.

Management for anaphylactic reaction (n = 23) included single actions of administering adrenaline (n = 2/23; 8.7%), calling an ambulance (n = 2/23; 8.7%) and laying the patient down (n = 1/23; 4.3%). Combinations of actions are shown in Table 5. Pharmacists undertook all three actions (i.e., managed correctly) in n = 13/23 (56.5%) of anaphylactic reaction cases. None of the pre-defined actions were taken in n = 2/23 (8.7%) cases, with one case (MMM 5) administering prednisolone before and after vaccination for their “usual” anaphylactic reaction symptoms and the other (MMM 1) “ended up in hospital” (this was coded as medical attendance). Other management actions for anaphylactic reaction (of which multiple could be undertaken) included self-monitoring for worsening symptoms at home (n = 1/23; 4.3%), continue monitoring in pharmacy (n = 1/23; 4.3%), giving water (n = 1/23; 4.3%), giving antihistamine (n = 1/23; 4.3%), checking blood pressure (n = 1/23; 4.3%), checking pulse (n = 1;23; 4.3%), following up patient the next day (n = 1/23; 4.3%), sitting patient down (n = 1/23; 4.3%), and monitoring breathing (n = 1/23; 4.3%). No anaphylactic reaction cases were medically referred and a total n = 18/23 (78.3%) were medically attended.

Pharmacists managed seizures (n = 23) primarily by laying the patient down (n = 7/23; 30.4%), calling an ambulance (n = 3/23; 13.0%), or both (n = 12/23; 52.2%). Overall, a total of n = 1/23 (4.3%) of seizures were medically referred, while n = 18/23 (78.3%) were medically attended (Table 6). Other management actions for seizure (of which multiple could be undertaken) were giving water (n = 4/23; 17.4%), giving glucose, lollipop, or chocolate (n = 3/23; 13.0%), monitoring in the pharmacy until symptoms resolved (n = 2/23; 8.7%), checking blood pressure (n = 2/23; 8.7%), elevating legs (n = 2/23; 8.7%), self-monitoring for worsening symptoms at home (n = 1/23; 4.3%), checking pulse (n = 1/23; 4.3%), and following up with the patient at the end of the day (n = 1/23; 4.3%).

Thematic analysis of ‘other’ management as recorded in free text entries, revealed 24 additional management actions taken by pharmacists of which some were administered, to varying degrees, alongside pre-defined management actions for the key I-AEFI anaphylactic reaction, syncope, nausea/vomiting, and seizure (as shown above). Non-pharmacological management included giving water, juice, electrolytes, glucose, lollipop, chocolate, or a cold compress; elevating legs or sitting the patient down; checking blood pressure, pulse, oxygen saturation, temperature, or blood sugar levels; and monitoring breathing. Pharmacological managements included giving salbutamol via a spacer, antihistamines, ibuprofen, paracetamol, or topical preparations (e.g., hydrocortisone); and advising to take prednisolone “as prescribed” for anaphylactic reaction (in this case the patient claimed their doctor had prescribed prednisolone as prophylaxis for vaccine-induced anaphylaxis). Pharmacists also recorded monitoring patients in the pharmacy until symptoms resolved; advising patients to self-monitor at home for worsening symptoms; providing government information on vaccines or consumer medicines information; reporting the I-AEFI to the Therapeutic Goods Administration; and following up with the patient at the end of the day. A full list of ‘other’ management actions taken by pharmacists in relation to the ‘other’ I-AEFI mapped to MedDRA^®^ is available in Appendix A.

## 4. Discussion

To our knowledge, this is the first study to investigate the types and management of immediate adverse events following immunisation (I-AEFI) following COVID-19 vaccination in community pharmacies. Results demonstrated that pharmacists are capable of identifying, recording, and managing a variety of different I-AEFI. The most common type of I-AEFI was syncope for all COVID-19 vaccine brands (AstraZeneca, Moderna, Novavax, Pfizer Paediatric and Pfizer). This is consistent with other studies investigating the types of I-AEFI experienced to COVID-19 vaccines [14,15,16]. The largest study monitoring I-AEFI, conducted in Italy, assessed I-AEFI occurring 15–120 min post-vaccination (AstraZeneca, Johnson & Johnson, Moderna, and Pfizer) at a mass vaccination hub [15]; 314,664 participants were vaccinated with 1409 developing an I-AEFI. The most common I-AEFI were a vagal response (30.0%), anxiety (24.1%), and dizziness (21.0%). An Australian study of I-AEFI following the Pfizer vaccination at a vaccination clinic, assessing I-AEFI occurring within 30 min post-vaccination, recorded 356 I-AEFI in 224 participants out of 57,842 vaccines administered [16]. The most reported I-AEFI were light-headedness (34.3%), nausea/emesis (14.9%), and headache (11.8%). The proportion of anyone experiencing any I-AEFI in the Italian (n = 1409/314,664; 0.45%) and Australian (n = 224/57,824; 0.39%) study were greater than reported in this study (n = 526/977,559; 0.05%), suggesting that pharmacists underreported I-AEFI.

Interestingly, the proportions of I-AEFI occurring were generally higher for the first dose when compared to the second or third+ doses. A systematic review on the safety of COVID-19 vaccines found that there were no statistically significant differences in the total number of adverse events following immunisation (AEFI), systemic adverse reactions, and local adverse reactions between the first and second dose [21]. Conversely, Cai et al. [22] found a higher incidence of AEFI following the second dose compared to the first. However, in both reviews, included studies recorded AEFI during clinical trials or from the Vaccine Adverse Event Reporting System (VAERS), which is a passive reporting system, and may result in overreporting or underreporting, respectively. Additionally, the clinical trials and VAERS include both immediate and non-immediate reactions. The differences observed in the proportion of I-AEFI by brand and dose in our study may be attributed to syncope or vasovagal syncope being triggered by stress [23,24]. It is possible that immunisation stress-related responses, including syncope [25], were more prominent during the early phases of vaccine rollout due to the fear of potential serious AEFI, which may have decreased with time and repeated vaccination as more information was available. This is apparent as in August 2020, in the early stages of the pandemic, 36% of an Australian sample expressed some level of hesitancy to COVID-19 vaccination [26], which may account for high rates of I-AEFI following the first dose if hesitancy decreased over time.

Of particular interest is the rate and proportion of anaphylactic reaction, as this is widely reported and a potentially life-threatening I-AEFI. Rates of anaphylactic reaction from EudraVigilance and VAERS were 67.7/10^6^ for AstraZeneca, 48.2–140.0/10^6^ for Moderna, and 195.9–360.2/10^6^ for Pfizer [27]. The overall approximate rates of anaphylactic reaction observed in this study were similar both for AstraZeneca (43.3/10^6^) and Moderna (29.2/10^6^) vaccines, but considerably lower for Pfizer vaccines (3.9/10^6^). Notably, only one case of anaphylaxis was recorded from over 250,000 Pfizer vaccinations in our study (the majority of which were third primary and booster doses). Furthermore, pharmacists could offer the Pfizer vaccine from November 2021, which was relatively late in the Pfizer arm of Australia’s vaccine rollout (which commenced in February 2021), and a large proportion of the population would have received the Pfizer vaccination elsewhere. Our low rates of anaphylaxis to the Pfizer vaccine may simply demonstrate that those people who had already experienced anaphylaxis to this vaccine in doses 1 or 2, were receiving third primary and booster doses at different locations or obtaining a different vaccine. The Australian study by Halder et al. [16] observed one case of anaphylaxis to the Pfizer vaccination from a sample of 34,300 people receiving their first dose, which gives an approximate rate of 29.2/10^6^. This rate is more similar to the rates observed to first doses of AstraZeneca and Moderna in our study. Notwithstanding, although the anaphylaxis rate reported by VAERS in the United States for the first dose of the Moderna vaccine (2.5/10^6^) are observed to be considerably lower than in this study [28], this rate is derived from a passive surveillance system and relies on the self-report of AEFI, which may result in underreporting. In comparison to the influenza vaccine, with anaphylactic reaction rates of 1.35/10^6^ and 1.83/10^6^ for the trivalent and monovalent vaccines, respectively, the rates reported in this study are considerably higher for COVID-19 vaccines [29]. However, this again may be due to differences in vaccine surveillance systems.

Overall, management of anaphylactic reaction by pharmacists was reasonable and appeared to be similar across metropolitan/regional and rural pharmacies, although there is room for improvement. The majority (56.5%) of anaphylactic reactions were managed correctly, whereby pharmacists undertook all three actions of administering adrenaline, laying the patient down, and calling an ambulance. Not laying the patient down is associated with poorer outcomes and increased risk of death [30]. Therefore, it is encouraging that in 70% of reactions, pharmacists laid the patient down. Similarly, a very high percentage of reactions (almost 80%) were treated with adrenaline, either alone or in combination with other actions. Ambulance transfer to hospital is recommended for all anaphylaxis events in the community [31], and this was undertaken in 70% of reactions. While this is lower than the proportion of cases that received adrenaline, this may be accounted for by the pharmacist physically transferring the patient to a nearby GP practice, and the outcome of the transfer was not recorded by the pharmacist, or the patient elected to take themselves to the emergency department. Our results suggest, that although approximately 20% of patients did not receive adrenaline (as recorded by the pharmacist), pharmacists have improved in administering adrenaline under emergency conditions, compared to previous research, which showed poor adrenaline autoinjector technique [32]. Furthermore, it is possible adrenaline was administered by paramedics or other medical professionals in the pharmacy, but not recorded by the pharmacist, as there was a high proportion of reactions that were medically attended. In context, the percentage of cases where adrenaline was administered by pharmacists was better than that observed by junior medical officers, where only 50% administered adrenaline in a simulated child presentation of definite anaphylactic reaction [33]. Results also showed that further training and more experience made the administration of adrenaline more likely [33]. It may be that as pharmacists have become more experienced delivering vaccinations (particular with high vaccination rates during the COVID-19 pandemic), they had gained confidence in recognising anaphylaxis and administering adrenaline.

The majority of syncope cases were managed by lying the patient down, which is appropriate in the management of vasovagal syncope as this may avert or attenuate syncope or traumatic falls [34]. The only obvious difference in the management of syncope by MMM appeared to be the additional management described in the free text, where monitoring in the pharmacy, and giving water were done in a higher percentage of cases by rural pharmacists. Interestingly, three cases of syncope were managed by administering adrenaline. While it is difficult to ascertain whether this management was appropriate based on the information provided in the dataset (i.e., there may have been other symptoms not recorded), this highlighted the possibility of cases of probable anaphylactic reaction that may be present in the dataset. A common way of identifying anaphylactic reaction from clinical records is the application of the Brighton Collaboration criteria for anaphylaxis [35]. To meet the criteria for anaphylactic reaction, sudden onset and rapid progression of signs and symptoms must be present; additionally, signs and/or symptoms from at least two body systems (skin, respiratory, cardiovascular, gastrointestinal, or laboratory) must be present. While anaphylactic reaction could be selected on the I-AEFI recording form, the specific clinical signs and symptoms could not be verified unless written in the free text. Including additional anaphylaxis symptoms, which can be identified by a pharmacist (e.g., skin reactions, persistent dry cough, and vomiting) alongside the anaphylactic reaction tick box in the I-AEFI recording form, would provide more reliable and verifiable anaphylactic reaction rates. Additionally, while pharmacists reported in free text that there were cases of seizures, it is impossible to ascertain whether these were true seizures, or cases of convulsive syncope, which may be mistaken as a seizure [36]. The number of jerks (<10 syncope; >20 seizure) and loss of tone (favours syncope), may help to differentiate between syncope and seizure [36].

## 5. Limitations

Although the I-AEFI recording form was available in all pharmacies using the MedAdvisor PlusOne software and pharmacists were advised to record I-AEFI as part of best practice, there was no way to ensure that all I-AEFI were being recorded. While this risk of underreporting was partially mitigated by only including pharmacies that had recorded at least one I-AEFI using the recording form, it cannot be confirmed whether all I-AEFI were recorded. Indeed, by including only pharmacies where I-AEFI were recorded, our results may overstate immediate reactions to COVID-19 vaccines (as a form of selection bias). Equally, lack of completion of the recording form by pharmacists when I-AEFI did occur, may have understated the true effects, had those pharmacies been included in the analysis. Additionally, patients may not have self-reported more minor I-AEFI (such as nausea or vomiting), or they may have left the pharmacy before the 15–30 min post-vaccination monitoring period ended, and any such reactions would have been missed.

We did not verify I-AEFI reactions; therefore, diagnoses remain those chosen by the pharmacist; however, our results demonstrate real-world identification and management of I-AEFI, and as such give insight into management practices and areas for improvement.

## 6. Conclusions

Immediate adverse events following immunisation (I-AEFI) occurred to AstraZeneca, Moderna, Novavax, and Pfizer COVID-19 vaccines; the most common I-AEFI was syncope. The approximate rate of anaphylactic reaction to AstraZeneca, Moderna and Pfizer vaccines ranged from 3.9–43.3/10^6^. The correct management of anaphylactic reaction (all three of: administered adrenaline, laid the patient down, and called an ambulance) was provided in 56.5% of cases, although almost 80% of pharmacists administered adrenaline.

Overall, pharmacists provided comprehensive care for patients experiencing I-AEFI, including providing non-pharmacological care, emergency care, medications, and monitoring and follow-up advice. To enhance the accuracy of future data, additional information in the I-AEFI recording systems should be implemented to provide a more complete clinical picture of I-AEFI, including symptoms that can be verified against Brighton Collaboration criteria.

## Figures and Tables

**Table 1 vaccines-10-02041-t001:** Demographics of participants with an I-AEFI to a COVID-19 vaccination by count.

		AstraZeneca (n = 161,857)	Moderna (n = 513,974)	Novavax (n = 11,967)	Pfizer Paediatric (n = 30,595)	Pfizer (n = 258,606)
**I-AEFI**	Any I-AEFI (proportion)	173	290	6	7	50
		(0.11%)	(0.06%)	(0.05%)	(0.02%)	(0.02%)
**Sex**	Female	67	147	4	4	24
	Male	106	142	2	3	26
	No sex recorded	0	1	0	0	0
**Age**	5–11	9	14	0	1	8
	12–15	0	0	0	4	0
	16–19	0	48	0	2	6
	20–29	20	32	1	0	2
	30–39	75	75	2	0	13
	40–49	33	51	1	0	12
	50–59	14	33	1	0	4
	60–69	11	19	1	0	3
	70–79	8	12	0	0	1
	80+	0	6	0	0	0
	No age recorded	3	0	0	0	1
**Australian State**	ACT	2	4	0	0	0
	NSW	107	47	1	2	15
	NT	0	2	0	0	0
	QLD	15	81	1	0	9
	SA	7	32	4	1	9
	TAS	0	20	0	0	4
	VIC	41	58	0	2	3
	WA	1	46	0	2	10
**Modified Monash Model** **Classification** **(MMM)**	1	130	181	5	3	41
	2	7	35	0	2	1
	3	10	27	0	0	4
	4	3	10	0	2	0
	5	23	33	1	0	4
	6	0	2	0	0	0
	7	0	2	0	0	0
**Recorded Dose**	First	155	209	4	6	23
	Second	18	45	2	1	8
	Third primary	0	1	0	0	3
	Booster	0	35	0	0	16

Note: No I-AEFI were recorded from 560 Moderna Paediatric vaccinations. ACT—Australian Capital Territory; I-AEFI—immediate adverse event following immunisation; MMM 1—metropolitan areas; MMM 2—regional centres; MMM 3—large rural towns; MMM 4—medium rural towns; MMM 5—small rural towns; MMM 6—remote communities; MMM 7—very remote communities; NSW—New South Wales; NT—Northern Territory; QLD—Queensland; SA—South Australia; TAS—Tasmania; VIC—Victoria; WA—Western Australia.

**Table 2 vaccines-10-02041-t002:** Count, proportion, and approximate rate per 1,000,000 vaccines administered of pre-defined I-AEFI and seizure experienced by participants by brand and dose of COVID-19 vaccines.

	AstraZeneca n = 161,857			Moderna n = 513,974				Pfizer n = 258,606				Pfizer Paediatric n = 30,595			Novavax n = 11,967	
	All Doses	1 n = 67,907	2 n = 89,919	All Doses	1 n = 108,339	2 n = 105,088	3 + n = 300,547	All Doses	1 n = 16,287	2 n = 21,095	3 + n = 221,224	All Doses	1 n = 16,544	2 n = 13,703	All Doses	1 n = 4665
**Syncope**	115	105	10	188	156	21	11	37	22	4	11	5	4	1	4	4
	(0.07%)	(0.15%)	(0.01%)	(0.04%)	(0.14%)	(0.02%)	(0.00%)	(0.01%)	(0.14%)	(0.02%)	(0.00%)	(0.02%)	(0.02%)	(0.01%)	(0.03%)	(0.09%)
	710.50	1546.23	111.21	365.78	1439.92	199.83	36.60	143.07	1350.77	189.62	49.72	163.43	241.78	72.98	334.25	857.45
**Nausea/vomiting**	36	31	5	73	55	12	6	14	9	2	3	3	3	0	0	0
	(0.02%)	(0.05%)	(0.01%)	(0.01%)	(0.05%)	(0.01%)	(0.00%)	(0.01%)	(0.06%)	(0.01%)	(0.00%)	(0.01%)	(0.02%)	-	-	-
	222.42	456.51	55.61	142.03	507.67	114.19	19.96	54.14	552.59	94.81	13.56	98.06	181.33	-	-	-
**Anaphylactic reaction**	7	6	1	15	9	1	5	1	0	1	0	0	0	0	0	0
	(0.00%)	(0.01%)	(0.00%)	(0.00%)	(0.01%)	(0.00%)	(0.00%)	(0.00%)	-	(0.00%)	-	-	-	-	-	-
	43.25	88.36	11.12	29.18	83.07	9.52	16.64	3.87	-	47.40	-	-	-	-	-	-
**Seizure**	4	4	0	15	12	1	2	3	2	0	1	1	1	0	0	0
	(0.00%)	(0.01%)	-	(0.00%)	(0.01%)	(0.00%)	(0.00%)	(0.00%)	(0.01%)	-	(0.00%)	(0.00%)	(0.01%)	-	-	-
	24.71	58.90	-	29.18	110.76	9.52	6.65	11.60	122.80	-	4.52	32.30	60.44	-	-	-

Notes: 1) Data presented as count, proportion of I-AEFI ((I-AEFI count/number of vaccines administered by brand and dose) × 100), and approximate rate of I-AEFI per 1,000,000 vaccines administered. 2) There were no pre-defined I-AEFI or seizure recorded from n = 560 Moderna Paediatric vaccinations, n = 4031 AstraZeneca dose 3+ vaccinations, n = 348 Pfizer Paediatric dose 3+ vaccinations, or n = 4430 Novavax dose 2 and n = 2872 Novavax dose 3+ vaccinations. 3) Dose 3+ represents all COVID-19 vaccinations administered after the standard 2-dose schedule (in this case, third primary dose and booster vaccinations are combined).

**Table 3 vaccines-10-02041-t003:** I-AEFI recorded as ‘other’ and mapped to MedDRA^®^ terminology preferred term.

Preferred Term	Count	Preferred Term	Count
Hyperhidrosis	20	Asthma	1
Injection site bleeding	17	Cardiac flutter	1
Paraesthesia	14	Cold sweat	1
Tremor	9	Confusional state	1
Pain in extremity	8	Cyanosis	1
Malaise	6	Dyspepsia	1
Tachycardia	6	Dysphonia	1
Feeling hot	5	Dystonia	1
Influenza like illness	5	Erythema	1
Rash	5	Feeling abnormal	1
Vaccination site rash	5	Foaming at mouth	1
Vision blurred	5	Heart rate decreased	1
Anxiety	4	Hot flush	1
Fatigue	4	Hyperventilation	1
Hypotension	4	Hypoaesthesia oral	1
Injection site mass	4	Limb discomfort	1
Tinnitus	4	Muscle tightness	1
Dysgeusia	3	Muscle twitching	1
Headache	3	Muscular weakness	1
Hypoaesthesia	3	Myalgia	1
Muscle spasms	3	Neuralgia	1
Paraesthesia oral	3	Orthostatic hypotension	1
Pyrexia	3	Pharyngeal hypoaesthesia	1
Angina pectoris	2	Pruritus	1
Chest pain	2	Respiratory arrest	1
Dyspnoea	2	Sensation of foreign body	1
Hypertension	2	Swelling	1
Injection site bruising	2	Swollen tongue	1
Lethargy	2	Urticaria	1
Palpitations	2		

**Table 4 vaccines-10-02041-t004:** Management of syncope occurring in the 15–30 min monitoring period after COVID-19 vaccination in community pharmacies.

MMM Classification	Called Ambulance Only n (%)	Laid Down Only n (%)	Laid Down and Called Ambulance n (%)	Gave Water n (%)	Monitored in Pharmacy until Symptoms Resolved n (%)	Gave Glucose, Lollipop, or Chocolate n (%)	Checked Blood Pressure n (%)	Medical Referral n (%)	Medical Attendance n (%)
**1–2 (MP/RG)**	15	184	43	44	33	19	24	3	66
**(n = 275)**	(5.5%)	(66.9%)	(15.6%)	(16.0%)	(12.0%)	(6.9%)	(8.7%)	(1.1%)	(24.0%)
**3–5 (rural)**	2	42	9	20	18	9	5	1	16
**(n = 66)**	(3.0%)	(63.6%)	(13.6%)	(30.3%)	(27.3%)	(13.6%)	(7.6%)	(1.5%)	(24.2%)
**6–7 (remote)**	0	1	0	0	0	0	0	0	0
**(n = 1)**	(0.0%)	(100%)	(0.0%)	(0.0%)	(0.0%)	(0.0%)	(0.0%)	(0.0%)	(0.0%)
**Total**	17	227	52	64	51	28	29	4	82
**(n = 342)**	(5.0%)	(66.4%)	(15.2%)	(18.7%)	(14.9%)	(8.2%)	(8.5%)	(1.2%)	(24.0%)

Note: Pharmacists could administer more than one type of management; percentages add to more than 100%. Management is presented as actions taken by pharmacists, in pharmacies based on Modified Monash Model groupings. Management of syncope did not include those that were also considered as anaphylactic reaction. MMM—Modified Monash Model; MP—metropolitan; RG—regional.

**Table 5 vaccines-10-02041-t005:** Management of anaphylactic reaction occurring in the 15–30 min monitoring period after COVID-19 vaccination in community pharmacies.

MMM Classification	Adrenaline Only n (%)	Called Ambulance Only n (%)	Laid Down Only n (%)	Adrenaline and Ambulance n (%)	Adrenaline and Laid Down n (%)	Managed Correctly ^a^ n (%)	Medical Attendance n (%)
**1–2 (MP/RG)**	1	1	1	0	2	8	10
**(n = 14)**	(7.1%)	(7.1%)	(7.1%)	(0.0%)	(14.3%)	(57.1%)	(71.4%)
**3–5 (rural)**	1	0	0	1	0	5	7
**(n = 8)**	(12.5%)	(0.0%)	(0.0%)	(12.5%)	(0.0%)	(62.5%)	(87.5%)
**6–7 (remote)**	0	1	0	0	0	0	1
**(n = 1)**	(0.0%)	(100%)	(0.0%)	(0.0%)	(0.0%)	(0.0%)	(100%)
**Total**	2	2	1	1	2	13	18
**(n = 23)**	(8.7%)	(8.7%)	(4.3%)	(4.3%)	(8.7%)	(56.5%)	(78.3%)

^a^ Managed correctly = administered adrenaline, laid the patient down, and called an ambulance. Management is presented as actions taken by pharmacists, in pharmacies based on Modified Monash Model groupings. There were no medical referrals or combinations of called ambulance and laid down. MMM—Modified Monash Model; MP—metropolitan; RG—regional.

**Table 6 vaccines-10-02041-t006:** Management of seizure occurring in the 15–30 min monitoring period after COVID-19 vaccination in community pharmacies.

MMM Classification	Called Ambulance Only n (%)	Laid Down Only n (%)	Called Ambulance and Laid Down n (%)	Medical Referral n (%)	Medical Attendance n (%)
**1–2 (MP/RG)**	2	4	11	1	15
**(n = 17)**	(11.8%)	(23.5%)	(64.7%)	(5.9%)	(88.2%)
**3–5 (rural)**	1	3	1	0	3
**(n = 6)**	(16.7%)	(50.0%)	(16.7%)	(0.0%)	(50%)
**6–7 (remote)**	-	-	-	-	-
**(n = 0)**	-	-	-	-	-
**Total**	3	7	12	1	18
**(n = 23)**	(13.0%)	(30.4%)	(52.2%)	(4.3%)	(78.3%)

Management is presented as actions taken by pharmacists, in pharmacies based on Modified Monash Model groupings. There were no cases recorded in MMM 6–7. MMM—Modified Monash Model; MP—metropolitan; RG—regional.

## Data Availability

Not available as we did not seek ethical approval for the release/sharing of data.

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
