# Peer review of "Immediate Adverse Events Following COVID-19 Vaccination in Australian Pharmacies: A Retrospective Review"

_vaccines, 2022, doi:10.3390/vaccines10122041_

Round 1

Reviewer 1 Report

The article entitled "Immediate Adverse Events Following COVID-19 Vaccination in Australian Pharmacies: A Retrospective Review" is fascinating work. Undoubtedly, the study carries great importance while many nations are providing booster doses amid the emergence of variants.

I highly appreciate the methodology of the work by which the study was conducted, specifically where the authors kept a clear discrimination between Medical referral and Medical attendance. 

However, there are some suggestions from my side which must be considered before the publication.

Abstract: In the conclusion, I suggest mentioning the Most commonly recorded I-AEFI is .... , then continue.

Line no. 32 to 33: Elaborate on this and make it clearer to read.

The emergence of variants and their association with the need for booster doses can be provided here. The articles mentioned below:

https://doi.org/10.1128/JVI.01973-21 

https://doi.org/10.1016/j.amsu.2022.103612

Line no. 129 to 137: I think if authors can elaborate on this more to make it more transparent, as this is the most crucial factor of the study. 

Additionally, My primary concern is why the authors did not only consider including only those subjects who were getting the first dose of the vaccine, as it is clear that there is a significantly high number of first-dose receipts in the study.

Hence, it is statistically not correct to compare the I-AEFI in two different groups of the population. 

As in the discussion, you have mentioned:

From lines 350 to 356: Interestingly, the proportions of I-AEFI occurring were generally higher for the first dose when compared to the second or third+ doses. A systematic review on the safety of COVID-19 vaccines found that there were no statistically significant differences in total adverse events following immunisation (AEFI), systemic adverse reactions, and local adverse reactions between the first and second dose [19]. Conversely, Cai et al. [20] found a higher incidence of AEFI following the second dose compared to the first. However, in both studies, AEFI included both immediate and delayed reactions. 

I request the authors to discuss this more and provide more information regarding the selection of the subjects and how this is justified with the present research work.

In the conclusion authors mentioned:

The approximate rate of anaphylactic reaction to AstraZeneca, Moderna and Pfizer vaccines ranged from 3.9–43.3/106 vaccines administered. 

I think it is statistically very low; I think the authors can mention something about vaccine hesitancy.

This study provides clear evidence that the COVID-19 vaccines are safe to administer, and there is no need to hesitate to get the COVID-19 vaccine and booster doses. 

https://doi.org/10.3390/vaccines10071155

In the end, I appreciate the efforts of the authors. 

Best Wishes

Author Response

Dear Reviewer, 

Thank you for taking the time to review this manuscript, which has made it of a higher standard. I have made most changes suggested or left comments regarding my choices.  

Abstract: In the conclusion, I suggest mentioning the Most commonly recorded I-AEFI is .... , then continue. Done

Line no. 32 to 33: Elaborate on this and make it clearer to read. Done

The emergence of variants and their association with the need for booster doses can be provided here.  Done

Line no. 129 to 137: I think if authors can elaborate on this more to make it more transparent, as this is the most crucial factor of the study. Done

Additionally, My primary concern is why the authors did not only consider including only those subjects who were getting the first dose of the vaccine, as it is clear that there is a significantly high number of first-dose receipts in the study. I am not sure I fully understood this comment; however, we wanted to investigate whether I-AEFI changed with the different doses. If we only looked at the first dose we would only see results when vaccine hesitancy was high and couldn't make any suggestions on how that might have changed over time. Also, by the time Pfizer was available in pharmacy, most people had already received their 1st and 2nd doses, so we would have lost a lot of participants only looking at 1st dose. Additionally, we did not make any statistical comparisons between 1st, 2nd, and 3rd+ doses so I feel this is ok; rather, we just reported observed proportions and estimated a rate.

From lines 350 to 356: Interestingly, the proportions of I-AEFI occurring were generally higher for the first dose when compared to the second or third+ doses. A systematic review on the safety of COVID-19 vaccines found that there were no statistically significant differences in total adverse events following immunisation (AEFI), systemic adverse reactions, and local adverse reactions between the first and second dose [19]. Conversely, Cai et al. [20] found a higher incidence of AEFI following the second dose compared to the first. However, in both studies, AEFI included both immediate and delayed reactions. I request the authors to discuss this more and provide more information regarding the selection of the subjects and how this is justified with the present research work. Done.

In the conclusion authors mentioned:

The approximate rate of anaphylactic reaction to AstraZeneca, Moderna and Pfizer vaccines ranged from 3.9–43.3/106 vaccines administered. I think it is statistically very low; I think the authors can mention something about vaccine hesitancy. This study provides clear evidence that the COVID-19 vaccines are safe to administer, and there is no need to hesitate to get the COVID-19 vaccine and booster doses. While I personally agree with this statement, since the aim of the study was not to determine the safety of COVID-19 vaccines, nor did we report non-immediate AEFI, I do not feel comfortable drawing this conclusion as hesitancy is about more than just the immediate reactions. 

Reviewer 2 Report

The manuscript submitted by Gallo et al and entitled “Immediate Adverse Events Following COVID-19 Vaccination in Australian Pharmacies: A Retrospective Review” describes a study that evaluated the effects of adverse events from 4 different COVID-19 vaccinations in an Australian pharmacy population.  In this review, all AEs were based on events that occurred 15-30 minutes post vaccination.  Overall, a total of 2248 pharmacies and almost 100,000 vaccinations performed.  Most common AE were syncope, nausea, anaphylactic reactions, and seizures.  AstraZeneca and Moderna had the most AEs in total, but overall rates were low ranging from 0.14% to 0.01%.  Overall, the paper is well written there are just some minor comments that are required prior to publication.

Major Comments

In the discussion it would be interesting to report the I-AEFI based on studies that have looked at other vaccines such as influenza.  This would allow a nice comparison to evaluate COVID-19 mRNA vaccines to other vaccination types.

Minor Comments

Overall, the manuscript is a bit long.  I might suggest reducing some of the discussion that talks about the management as that is mostly repeated from the results section.

Author Response

Dear Reviewer, 

Thank you for taking the time to review this manuscript. I have made the changes requested. 

Major Comments

In the discussion it would be interesting to report the I-AEFI based on studies that have looked at other vaccines such as influenza.  This would allow a nice comparison to evaluate COVID-19 mRNA vaccines to other vaccination types. Included in discussion. 

Minor Comments

Overall, the manuscript is a bit long.  I might suggest reducing some of the discussion that talks about the management as that is mostly repeated from the results section. Deleted paragraph in discussion about management by MMM as this was, as you said, a repeat of the results.

Round 2

Reviewer 1 Report

The authors have incorporated all the required information and updated the manuscript as per the reviewers comments.  

Minor correction: The virus came to be known as the coronavirus disease 2019 (COVID-19), COVID-19 is not the viral name, Check the cited reference.  1 or 3 and change it appropriately. 

Best Wishes 

Author Response

Thank you! Corrected.